# A background correction method to compensate illumination variation in hyperspectral imaging

Jonghee Yoon[1,2]*, Alexandru Grigoroiu[1,2], Sarah E. Bohndiek[1,2]*

1 Department of Physics, University of Cambridge, Cambridge, England, United Kingdom, 2 Li Ka Shing Centre, Cancer Research UK Cambridge Institute, University of Cambridge, Cambridge, England, United Kingdom

* jonghee.yoon@cruk.cam.ac.uk (JY); seb53@cam.ac.uk (SEB)

**Data Availability Statement:** Data will be uploaded into the University of Cambridge Open data repository Apollo, more information can be found at: www.data.cam.ac.uk. The DOI for this data set is https://doi.org/10.17863/CAM.42338

## Abstract

Hyperspectral imaging (HSI) can measure both spatial (morphological) and spectral (biochemical) information from biological tissues. While HSI appears promising for biomedical applications, interpretation of hyperspectral images can be challenging when data is acquired in complex biological environments. Variations in surface topology or optical power distribution at the sample, encountered for example during endoscopy, can lead to errors in post-processing of the HSI data, compromising disease diagnostic capabilities. Here, we propose a background correction method to compensate for such variations, which estimates the optical properties of illumination at the target based on the normalised spectral profile of the light source and the measured HSI intensity values at a fixed wavelength where the absorption characteristics of the sample are relatively low (in this case, 800 nm). We demonstrate the feasibility of the proposed method by imaging blood samples, tissue-mimicking phantoms, and *ex vivo* chicken tissue. Moreover, using synthetic HSI data composed from experimentally measured spectra, we show the proposed method would improve statistical analysis of HSI data. The proposed method could help the implementation of HSI techniques in practical clinical applications, where controlling the illumination pattern and power is difficult.

## Introduction

Hyperspectral imaging, originating from remote sensing applications[1], enables a combined simultaneous measurement of both spatial and spectral information from biological tissues. Analysis of the resulting 3D data set, or 'hypercube', enables spatial discrimination of healthy and abnormal tissues based on the rich morphological and biochemical information contained within the spatial and spectral features[2,3]. HSI has shown potential in a range of biomedical applications, from label-free tumour diagnoses[4–6] and detection of tumour margins during surgical operations[7–9], to quantification of blood oxygenation levels[10–12], and multi-colour fluorescent imaging[12,13]. HSI methods have thus been developed for the fast and accurate analysis of biological samples *ex vivo*[14–17] as well as for diagnostic and intraoperative applications *in vivo*[16,18].

**Funding:** This work was funded by CRUK (C47594/
A16267, C14303/A17197, C9545/A29580,
C47594/A21102; www.cancerresearchuk.org),
EPSRC (EP/R003599/1, EP/N014588/1, EP/
L015889/1; www.epsrc.ac.uk) and the EU FP7
agreement FP7-PEOPLE-2013-CIG-630729 (ec.
europa.eu). These grants were awarded to SEB and
support the work of JY and AG, with the exception
of EP/L015889/1, which is a Centre for Doctoral
Training award that supports AG. The funders had
no role in study design, data collection and
analysis, decision to publish, or preparation of the
manuscript.

**Competing interests:** The authors have declared
that no competing interests exist.

The high complexity of handling the 3D hypercube requires careful consideration of appropriate analysis methods[3,19–21]. HSI data are commonly subjected to a normalisation procedure to calculate reflectance and/or absorbance of the sample using the following Eq[3]:

$$R(x, y, \lambda) = \frac{I(x, y, \lambda) - I_{dark}}{I_0(x, y, \lambda) - I_{dark}} \tag{1}$$

$$A(x, y, \lambda) = -log_{10}\left(\frac{I(x, y, \lambda) - I_{dark}}{I_0(x, y, \lambda) - I_{dark}}\right) \tag{2}$$

,where $R(x,y,\lambda)$ and $A(x,y,\lambda)$ are the reflectance and absorbance at a given spatial position $(x, y)$ and wavelength $(\lambda)$, respectively. $I$, $I_0$, and $I_{dark}$ are the intensities of the spectral signals measured from the sample, the background spectral signals recorded without the sample in place (also referred to as the "white" signals) and the dark signals recorded without any illumination, respectively. Reflectance and absorbance metrics thus indicate the true spectral features of a sample as these calculations correct for variations in illumination conditions and errors introduced by the optical components. These processed reflectance or absorbance signals can then be further subjected to statistical analyses such as principal component analysis (PCA)[22,23], spectral angle mapping (SAM)[24], or machine learning methods[5,25–27], to extract significant spectral features that can discriminate or classify the samples of interest.

A key limitation of these reflectance and absorbance calculations is the assumption of uniform sample illumination. Several methods are used in hardware to ensure that this assumption remains valid, including: uniform illumination instrumentation[28,29]; 3D shape measurement[30]; reference intrinsic / fluorescence imaging[31–34]; and ratiometric measurements[35–37]. Uniform illumination instrumentation can be achieved with specialised devices such as ring illuminators or diffuse domes, however, there are difficulties in applying these for *in vivo* imaging, such as during endoscopy, due to the need for bulky illuminating units. In addition, uniform illumination instrumentation does not guarantee intensity homogeneity along the axial direction, which means illumination issues can still occur *ex vivo* when measuring most biological tissues due to their uneven surfaces. Estimating optical illumination power is possible using 3D shape measurement techniques and optical model-based analysis but predicting illumination conditions within shadowed regions is challenging. Intrinsic image or reference fluorescence signals measured by multimodal imaging systems have been used to provide a reference background (BG) that enables estimation of the optical power distribution of the light source and correction of sample signals but again, a complex optical system is required to measure reference BGs and additional errors are introduced by variations in tissue absorption or the concentration of fluorescence agents. Ratiometric measurements, such as narrow-band imaging, record spectral information from only a few spectral bands, displaying physiological information based on a (weighted) sum of the images. Although ratiometric imaging is usually insensitive to illumination conditions and sample morphology, only limited spectral information is recorded.

Here, we introduce a BG correction method that estimates the optical power of illumination at a sample by exploiting the normalised spectral profile of the light source and the hyperspectral signal of the sample. We experimentally demonstrate the proof-of-concept of the method using HSI data acquired via a hyperspectral endoscopy system from blood samples, tissue-mimicking phantoms, and *ex vivo* chicken tissue. Moreover, the importance and applicability of the proposed method to hyperspectral image analysis (PCA and SAM) and machine learning classification of hyperspectral data were tested using synthetic reflection and absorption hypercubes based on these experimentally measured spectra. The proposed BG correction

method, referred to as retrieved background (RB), enables the estimation of optical characteristics of illumination at the sample, avoiding the need for additional complex hardware, and results in accurate hyperspectral data analysis and classification.

## Materials and methods

### Hyperspectral imaging system

The hyperspectral imaging endoscopy system, reported previously[18], exploits a flexible CE-marked endoscope (Polyscope, PolyDiagnostics) and a line-scanning (pushbroom) method. Briefly, the endoscope consists of a reusable imaging fibre bundle with 10,000 individual fibrelets and a disposable sterile catheter that contains an imaging channel, an illumination fibre and an accessory channel. The proximal end of the imaging fibre bundle was imaged and magnified using an infinity corrected objective lens (40×, 0.6NA, Nikon) and a tube lens (L1, f = 75 mm), with the image being measured by an electron multiplying CCD camera (sCam, ProEM 512, Princeton Instruments) combined with a spectrograph (IsoPlane 160, Princeton Instrument) to obtain hyperspectral information. The spectrograph consisted of a mechanical entrance slit of manually adjustable width (10 μm– 3 mm) and a grating (150 lines/mm with 500 nm blaze, Princeton Instruments); thus a spectral image with a spectral bandwidth of 250 nm can be measured in a single image acquisition. The spectrograph and camera were controlled by LightField software v6.7 (Princeton Instrument). In order to obtain a wide-area hyperspectral image, the line-scanning was performed using a motorized translational stage (MTS50/M-Z8, Thorlabs). All equipment was synchronously controlled in Labview 2017 (National Instruments) environment.

A broadband light source (OSL2, Thorlabs) with a Halogen light bulb (OSL2bIR, Thorlabs) whose emission spectrum spanned across the visible to NIR (1050 nm) region was used to illuminate a sample either internally or externally, depending on experimental purposes. For internal illumination, the light source was directly coupled to the illumination fibre of the endoscope by using a collimating lens (L2, f = 150 mm) and an objective lens (60×, NA 0.9, Olympus). For external illumination, the light source was coupled to a large core fibre and the distal end of the fibre was placed 2 cm away from the sample at a tilted angle.

### Image acquisition

Spectral image acquisition was performed after allowing 15 minutes for temperature stabilisation of the equipment. The image acquisition process consisted of three steps: (1) dark imaging; (2) white reflectance imaging; and (3) sample measurement. Dark imaging was performed under closed camera shutter conditions. White reflectance imaging was performed using a standard white reflectance target (Spectralon diffuse reflectance target, Labsphere) to obtain information of the spectral profile and intensity of the light source. All image acquisition processes were performed under the same experimental conditions, including exposure time, gain and light source power.

### Hypercube reconstruction

The recorded 2D line-scan image contains one spatial coordinate and the spectral coordinate, because the grating inside the spectrograph disperses the image horizontally; hypercube reconstruction is required to retrieve the other spatial coordinate, obtained during the motorized translation. Before commencing hypercube reconstruction, the dark image was subtracted from the white reflectance and sample images. A single column of the corrected image, which contains information from a single wavelength, was selected and duplicated horizontally to

match its image size to the physical slit width. For example, in hyperspectral imaging of the chicken tissue, line-scan hyperspectral images of the sample and white-reflectance target were measured with a step scanning size of 250 μm, which corresponds to 5 pixels. Thus, each processed image was placed 5 pixels apart from the previous image. By repeating this process for all column images, a slice of the hypercube at a single wavelength was created. The 3D hypercube was then reconstructed by repeating the process to create a wide-area spatial image at all wavelengths. Hypercubes of the sample and white-reflectance target were reconstructed separately, enabling the calculation of normalised reflectance and absorbance values by dividing the sample and white-reflectance hypercubes.

## Generation of a synthetic RGB image from the hypercube

For visualization purposes, the hypercube can be converted to a synthetic RGB (colour) image using an artificially generated RGB filter based on a previously published method[18]. The spectrum of the RGB filter employed Rayleigh probability density functions (raylpdf function in Matlab R2018b), with centre wavelengths of each colour being set to 442, 518, and 579 nm, respectively. Amplitudes of each filter were determined such that saturation of the synthetic RGB image was avoided. The hyperspectral signal from the hypercube was multiplied by the artificially generated RGB filters, with the R, G and B values of the synthetic RGB image being determined by calculating the area-under-curve values of the filtered signals. Synthesized RGB images were displayed using imshow function in Matlab R2018b.

## Preparation of chicken tissue

A food-grade chicken drumstick purchased from a local grocery market was horizontally dissected using a knife. Local handling of the tissue was approved by our Biological Safety Committee. The test sample was then placed on a petri dish and measured by using the hyperspectral endoscope. To obtain background signals, a white-diffuse-reflectance target was measured under the same experimental conditions as the sample measurement. The hyperspectral imaging was performed at a working distance of 7 cm with a step size of 250 μm on the motorized stage. A total of 150 spectral images were measured, resulting in a total scanning area of 31.56 mm × 37.50 mm, with an exposure time of 1s. The experiments were conducted within a 3 hour timeframe to ensure sample freshness.

## Preparation of tissue-mimicking phantom and blood samples

For blood oxygenation measurements, fresh heparinized mouse blood was collected from deceased mice provided by the Biological Resources Unit of the Cancer Research UK Cambridge Institute (mice were not sacrificed for the purpose of this study). 1 mL of mouse blood was divided between two 1 mL Eppendorf tubes. To make a fully oxygenated blood sample, 1 μL of 30% hydrogen peroxide (Sigma-Aldrich) was added and the sample was gently mixed by inversion. 1.5 mg of sodium hydrosulphite (Sigma-Aldrich) was added to the other tube to make a completely deoxygenated blood sample, again mixing by inversion. The tubes were kept at room temperature for 10 mins and 20 μL of the oxygenated and deoxygenated blood samples were transferred to a petri dish and covered by a cover slip. As a reference target, 20 μL of distilled water was put on the petri dish and covered by the cover slip.

To test the effects of scattering, absorption and fluorescence on the suggested method, tissue mimicking phantoms with defined optical properties that closely mimic biological tissue were fabricated using agar, intralipid, nigrosin and methylene blue[38]. All chemicals were purchased from Sigma-Aldrich. Before fabricating the tissue phantoms, two different concentrations of absorbance and fluorescence dyes were prepared. Nigrosin (0.1 and 0.05 g/mL) and

methylene blue (0.1 and 0.05%) were prepared by diluting dyes using distilled water. 0.75 g of agarose was dissolved in 48.5 mL of distilled water and then heated to the boiling point using a microwave oven. The solution was left to cool to ~40˚C, with 1 mL of 20% intralipid being added to the solution and gently mixed to induce optical scattering. 500 μL of the solution was transferred to 6 wells of an 8 well dish (μ-Slide 8 Well, ibidi GmbH) using a pipette and then 100 μL of the four prepared dyes were added to 4 of the cells. The dish was covered by plastic wrap and kept inside a refrigerator to set.

## Creation of synthetic absorption and reflection hypercubes

In order to test machine-learning methods, synthetic hypercubes that mimic experimental conditions were used, composed of spectral signals from experimentally measured BG and samples (pork muscle tissue, oxygenated blood, methylene blue and nigrosin dyes). All samples were measured four times under different experimental conditions to include noise generated by the optical systems and environment to synthetic hypercubes. To generate uncorrelated training and test hypercubes, three of the four measured data sets were used for training data and the other data set was used for test data.

Synthetic reflection and absorption hypercubes were created by following four steps: (1) generation of a random illumination pattern; (2) creation of a GT reflectance hypercube based on four experimentally measured signals with an uncorrelated noise; (3) creation of SB and RB reflectance hypercubes by combining the GT hypercube with the random illumination pattern; and (4) applying a log-transformation of the produced reflectance hypercubes to generate absorbance hypercubes

Step (1): 2D random Gaussian distributions, $M$, were used as ground-truth optical power distributions, with values were ranging between 0 and 1. Gaussian distribution was created using 'mvnpdf' function in Matlab, and its central location was randomly assigned using 'rand' function in Matlab. Optical characteristics of the illumination conditions were decided by the following equation:

$$BG(x, y, \lambda) = M(x, y) \times S_{light}(\lambda)$$

where $BG(x,y,\lambda)$ is light intensity at the wavelength of $\lambda$ at the point $x,y$ in the image, $M(x,y)$ is the optical power at the point $x,y$, and $S_{light}(\lambda)$ is the experimentally measured spectral intensity of the light source at the wavelength of $\lambda$, respectively.

Step (2): The GT reflectance hypercube was created by assigning experimentally measured hyperspectral signals of samples (pork muscle tissue, oxygenated blood, methylene blue, and nigrosin dye) with an uncorrelated noise obtained from independent experimental measurements of spectral signals from a colour chart (ColorChecker Classic Mini, x-rite) to each of the corresponding clusters in the spatial regions of the image (either target or background regions).

$$GT(x, y, \lambda) = S_{sample}(\lambda) + \alpha \times N(\lambda)$$

where $S_{sample}(\lambda)$ is the experimentally measured spectral intensity among the samples at the wavelength of $\lambda$, $\alpha$ is randomly generated weighting factor between 0 to 0.1 ('rand' function in Matlab), and $N(\lambda)$ is an experimentally measured spectral intensity of the colour chart at the wavelength of $\lambda$, respectively. The noise, $\alpha \times N(\lambda)$ has a scale less than 10% of sample signals,

$S_{sample}(\lambda)$, and the range of the weighting factor, $\alpha$, was determined to make the noise scale consistent with the scale of average experimental noise. The uncorrelated noise makes the training process more robust and reduces generalization error.

Step (3): SB and RB reflectance hypercubes were created based on the GT hypercube from Step (2) and single and retrieved illumination conditions by following equations, respectively:

$$SB(x, y, \lambda) = \frac{BG(x, y, \lambda) \times GT(x, y, \lambda)}{S_{light}(\lambda)}$$

$$RB(x, y, \lambda) = \frac{BG(x, y, \lambda) \times GT(x, y, \lambda)}{RM(x, y, \lambda)}$$

where $RM(x,y,\lambda)$ is retrieved optical power at the wavelength of $\lambda$ at the point $x,y$ obtained via the BG retrieval method.

Step (4): Absorbance hypercubes were calculated by performing logarithmic transformation of the GT, SB and RB reflectance hypercubes.

## Principal component analysis

A pixel-wise approach and singular value decomposition (SVD) were exploited to perform PCA of the hypercube[39]. Three pre-processing steps were required before calculating the SVD of the hypercube. First, the 3D hypercube was vectorised into a 2D matrix, consisting of pixels (vertical axis) and hyperspectral signals (horizontal axis). Then, hyperspectral data was centred by subtracting mean values of the hyperspectral signal of each pixel from its corresponding signal. Finally, the covariance matrix of the pre-processed hyperspectral data was calculated, which was used as an input of SVD. SVD was performed using *svd* function in Matlab R2018b. An NVIDIA GeForce GTX 1080 graphical processing unit was exploited for fast SVD calculation.

## Specular angle mapping

For SAM analysis, the average hyperspectral signal of cluster $i$ of each hypercube was used as a reference hyperspectral signal. Then, the spectral angles, $\alpha$, between the hyperspectral signal of each pixel of a hypercube and the reference spectral signal were calculated using the following equation[24]:

$$\alpha = \cos^{-1}\left(\frac{\sum_{\lambda=1}^{n} t_\lambda r_\lambda}{\left(\sum_{\lambda=1}^{n} t_\lambda^2\right)^{0.5}\left(\sum_{\lambda=1}^{n} r_\lambda^2\right)^{0.5}}\right) \quad (3)$$

, where $t_\lambda$ and $r_\lambda$ are values of the target and reference spectral profiles at wavelength $\lambda$, respectively and $n$ indicates the total number of spectral channels.

## Machine-learning based classification of emulated hypercube

Learning algorithms were implemented in Python, with K-means clustering and SVM algorithms being implemented via the sklearn library and CNNs being implemented via Lasagne, a Theano supplementary library. Learning was performed on a machine with access to 16 GB RAM and a NVIDIA GeForce GTX 1050Ti graphical processing unit. To validate the lack of overfitting of the classifiers, the synthetic data was split into training and testing datasets, with

base spectral measurements for the simulations being independent of one-another. All results presented are based on the performance of the classifiers on the test data.

A pixel-wise approach was used for learning and classification processes. A 3D hypercube, consisting of 256 × 512 spatial points and 300 spectral channels, was converted to a 2D image with sizes of 131072 × 300. Each row of the converted image with 300 spectral channels was then used as an input to the learning and classification processes. We found that twenty-five training datasets were sufficient to achieve 100% classification accuracy in the ground-truth data. For a better supervised learning process, fifty hypercubes were exploited to train supervised learning models (SVMs and CNNs). Due to the large data sizes and memory limitations, learning was performed incrementally in batches of one hypercube (131072 × 300). Thus, for the K-mean algorithm, the MiniBatchKMeans function was employed, with a 21 epochs early stopping decision and a target of 4 clusters. An incremental SVM algorithm has been implemented by employing the SGDClassifier function with a hinge loss function and l2 regularisers. For the CNN a six layered network was implemented with three convolutional layers, two fully connected layers and an output layer. Unlike in the previous two methods (K-means and SVM algorithms), a subset of a hypercube (200 × 300) was used as a batch size of CNN to facilitate a more effective learning process. 5-fold cross-validation was performed to test the accuracy of CNNs.

## Software

Matlab R2017b and Python were used for image processing. Lightfield v6.7 (Princeton Instrument) was used to control the spectrograph and EMCCD. Labview 2017 (National Instruments) was used for synchronized control of the wide-field camera, spectrograph and EMCCD, and motorized stage.

## Code availability

All custom data analysis code will be made available online at: https://doi.org/10.17863/CAM.42338

## Results

### Background correction using the normalised source profile and target hyperspectral signals

The influence of varying illumination power on the calculation of reflectance and absorbance spectra along with the proposed correction method is demonstrated in Fig 1. Experimentally measured hyperspectral signals were acquired from absorbing nigrosin black dye as the sample (Fig 1A) and a standard diffuse reflectance target as the background (Fig 1B, $\omega = 1$). To emulate varying illumination intensities, two weighting factors were multiplied with the ground-truth BG signal (Fig 1B, $\omega = 0.8, 1.2$) and the resulting reflectance (Fig 1C) and absorbance (Fig 1D) spectra were calculated according to the following equations (see Methods for complete definition of all variables):

$$R(\lambda) = \frac{I(\lambda) - I_{dark}}{\omega \times (I_0(\lambda) - I_{dark})} \tag{4}$$

$$A(\lambda) = -log_{10}\left(\frac{I(\lambda) - I_{dark}}{\omega \times (I_0(\lambda) - I_{dark})}\right) = -log_{10}\left(\frac{I(\lambda) - I_{dark}}{(I_0(\lambda) - I_{dark})}\right) + log_{10}w \tag{5}$$

Varying the intensity in this way resulted in the expected change in the scale of the calculated

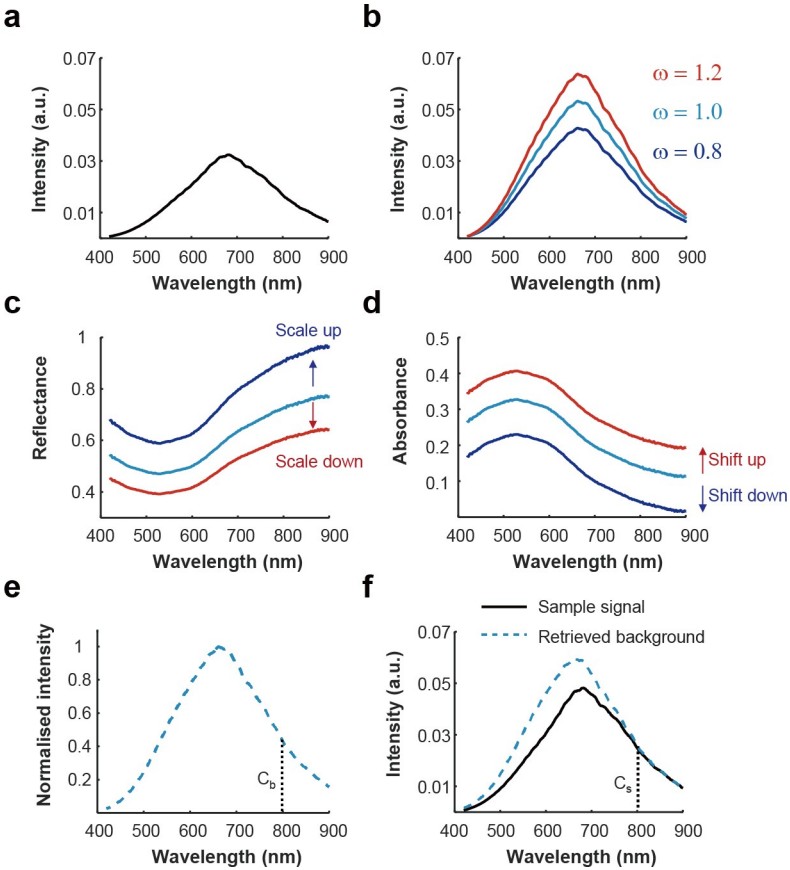

**Fig 1. The effect of illumination power on absorbance and reflectance spectra.** Raw spectra of 0.05 g/mL nigrosin dye **(a)** and the background halogen light source **(b, $\omega = 1$)** were measured. To simulate the effects of the low and high illumination power, weighting factors ($\omega = 0.8$ and 1.2) were multiplied with the background spectrum. **(c, d)** Reflectance and absorbance were obtained for the three different weighting factors. Illumination power can be observed to change the scaling of reflectance spectra and the offset of absorbance spectra. **(e, f)** The proposed background retrieval method estimates the optical spectral power of the illumination at the sample by exploiting a normalised spectral profile of the light source **(e)** and the intensity ratio between the normalised spectral profile of the light source and sample data ($C_s/C_b$) at a wavelength displaying low absorbance in the sample of interest, here selected as 800 nm **(f)**.

reflectance, since $\omega$ is a multiplicative factor in Eq (4), and a change in the offset of the calculated absorbance, because $\omega$ becomes an additive constant in Eq (5) due to logarithm calculations. This simple illustration highlights how image processing with incorrect BG data would cause errors in the interpretation of hyperspectral data. We propose instead to multiply the normalised spectrum of the light source (Fig 1E) with the intensity ratio between the normalised spectral profiles of the light source ($C_b$) and the sample ($C_s$) at a wavelength of low absorbance in the sample to estimate the actual spectrum of the light source at the target (Fig 1F). Here, we select 800 nm as the wavelength for comparison, since this is central in the near-infrared (NIR) tissue 'optical window' of low absorbance in biological tissues[40,41]. Moreover, there is an isosbestic absorption of haemoglobin at 800 nm meaning that any change in absorption due to haemoglobin oxygenation status would not affect the calculation and the loss of information at the normalisation is minimal since there are several other isosbestic points for haemoglobin within the visible spectral region.

The proposed method calculates reflectance and/or absorbance of the sample using the following equations:

$$R(x, y, \lambda) = \frac{I(x, y, \lambda) - I_{dark}}{\frac{Cs(x,y)}{Cb(x,y)} \times NS(\lambda)} \qquad (6)$$

$$A(x, y, \lambda) = -log_{10} \left( \frac{I(x, y, \lambda) - I_{dark}}{\frac{Cs(x,y)}{Cb(x,y)} \times NS(\lambda)} \right) \qquad (7)$$

, where $NS(\lambda)$ is the normalised spectrum of the light source, $Cs(x,y)$ and $Cb(x,y)$ are intensity values of $I(800)$ at the point $x,y$, and $NS(800)$, respectively.

## Proof-of-concept using a standard reflectance target and phantoms

In order to test the proposed method, hyperspectral imaging data were acquired via a hyperspectral endoscope (HySE) that consists of a line-scanning spectrograph and multi-core optical fibre endoscope (S1 Fig, see Methods)[18]. The endoscopy system can image the sample using light from an external fibre-coupled light source (referred to as 'external illumination') or using light delivered through an internal illumination fibre (referred to as 'internal illumination'). We introduce external illumination here to provide light with an easily adjustable distance and angle relative to the sample. During clinical endoscopy, internal illumination is used, and the changing working distance and angle of endoscope lead to additional heterogeneities in sample illumination.

Data were first acquired from a standard white reflectance target that reflects 99% of illuminating light using external illumination (Fig 2A), where a fibre coupled to a broadband light source was tilted to create a variation in the optical power distribution across the sample. Line-scanning HSI was performed at three different positions (indicated in Fig 2A with coloured rectangles) and the resulting line-scan HySE image containing 1D spatial (vertical axis) and spectral (horizontal axis) information (Fig 2B) was then processed at each position to retrieve the average spectral profiles (Fig 2C). Min-max normalisation led to complete overlap of the spectra (Fig 2D), indicating that the light source illuminates each position with the same spectral profile but with different optical powers. Applying the proposed BG correction method, ratios of the intensity values of three hyperspectral signals (Fig 2C; 2C$_1$, 2C$_2$ and 2C$_3$), and the normalised signal (Fig 2D and 2C$_b$) at 800 nm were taken ($c_1/c_b$, $c_2/c_b$, and $c_3/c_b$) and multiplied by the normalised spectrum (Fig 2D) to successfully retrieve the original signal (Fig 2E). Repeating the same process via internal illumination (Fig 2F) also showed appropriate background retrieval (Fig 2G–2J).

To compare our results to other BG correction methods when imaging a range of samples, we then defined three different BG conditions: ground-truth BG (GT); single BG (SB); and our retrieved BG method (RB). GT was obtained by measuring HSI data from the standard white reflectance target under precisely the same conditions as the sample imaging (S2A Fig). For example, GT data was acquired at every working distance used. This is rarely feasible in practical clinical applications, such as during endoscopy, as a reference target cannot be introduced into the lumen being imaged nor are working distance variations normally accounted for. SB is the conventional background correction method commonly used in HSI and obtained by measuring HSI data from the standard white reflectance target prior to sample measurements under arbitrary illumination conditions and assuming this spectral profile to be representative of the illumination conditions during the sample imaging[3]. SB does not allow compensation of any variations that are introduced during the imaging condition, such as variations

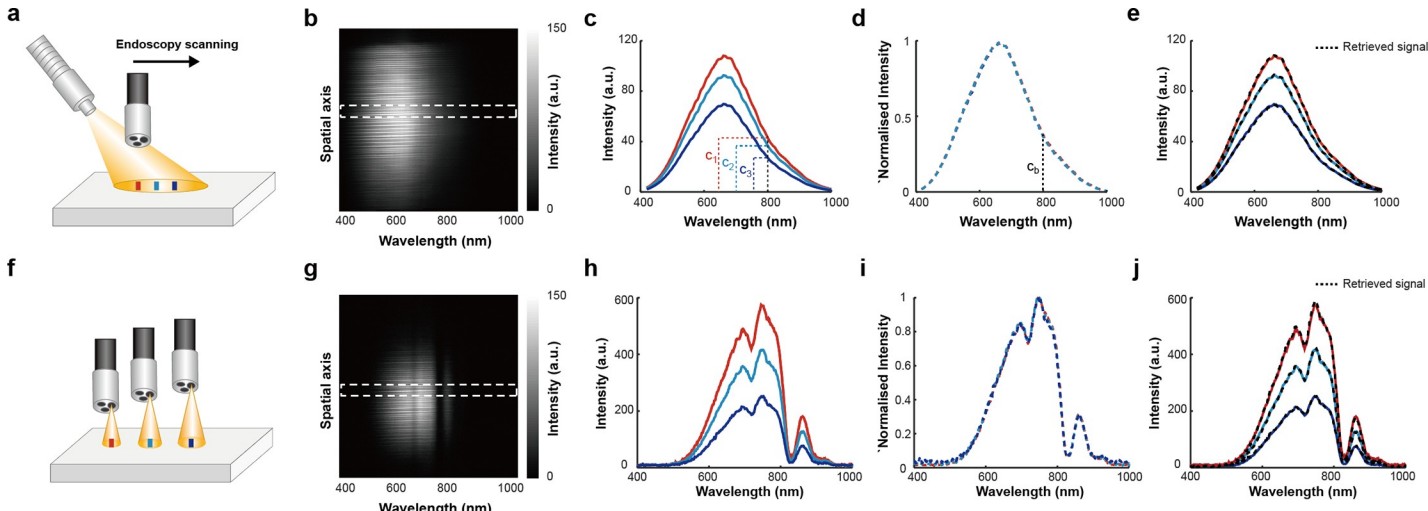

**Fig 2. Spectral profiles of different illumination conditions can be accurately retrieved through hyperspectral endoscopy by different illumination methods. (a)** Schematic of the external illumination methods. Red, green, and blue lines indicate the hyperspectral imaging regions. **(b)** Representative spectral image from the hyperspectral endoscope during external illumination. **(c)** Average spectral signals of the three hyperspectral images measured at the different locations shown in (a) obtained from the white-dashed region in (b). **(d)** Min-max normalisation of spectral signals in (c) show complete overlap. **(e)** Spectra obtained (black dashed lines) using the normalised spectral profile in (d) and ratio of values at 800 nm ($c_1$, $c_2$, $c_3$, and $c_b$). **(f-j)** As above but for the internal illumination method.

illumination power and working distance. In the present study, an arbitrary choice of a single GT HSI data sets was taken as SB. RB is calculated as described above (S2B and S2C Fig).

We then analysed blood samples, since blood has distinct absorption profiles depending on the level of oxygenation and provides relatively low absorbance at 800 nm[42]. Fully oxygenated and deoxygenated blood (see Methods) and distilled water (20 μL) were pipetted onto a plate, covered by a coverslip (Fig 3A) and imaged immediately. HySE was applied using internal illumination at 3 working distances (Fig 3B) and the spectral profile of distilled water was used as GT. The measured absorption spectra of the deoxygenated (Fig 3C) and oxygenated blood (Fig 3D) clearly show one (550 nm) and two (540 and 560 nm) peaks respectively, consistent with known blood absorption spectra[42]. The slightly different absorption values of three measurements might be originating from the varying imaging areas, which are dependent on the working distance of the endoscope. The absorbance spectra calculated using the retrieved BG are consistent with the GT results, albeit slightly lower in magnitude (8.73 ± 1.56% lower at 550 nm peak and 6.84 ± 1.22% lower at 560 nm peak of the absorption spectra of the deoxygenated and oxygenated blood, respectively), however, the conventional single BG method produces substantial differences.

As the proposed method only uses a single normalisation wavelength, the accuracy of the method may be affected by noise. The influence of noise levels to the retrieved signals was assessed via simulation (S3 Fig). Four different noise levels (1%, 5%, 10%, and 20%) were added to the spectral profile of oxygenated blood (S3A Fig) and absorbance was calculated based on simulated spectral signals with different noise levels and the proposed method (S3B Fig). S3C Fig shows that the error levels are 3.62 ± 0.31%, 4.35 ± 1.64, 4.98 ± 2.26%, and 7.81 ± 2.90% with increasing a noise level from 1% to 20%, respectively. This indicates that high noise levels could compromise the accuracy of the proposed method and care should be taken when applying the approach to noisy spectra.

To demonstrate that the BG correction method remains accurate under scattering, absorption and fluorescence conditions, a tissue-mimicking phantom was exploited. The phantom was made of agarose and intralipid with high and low concentrations of nigrosin and

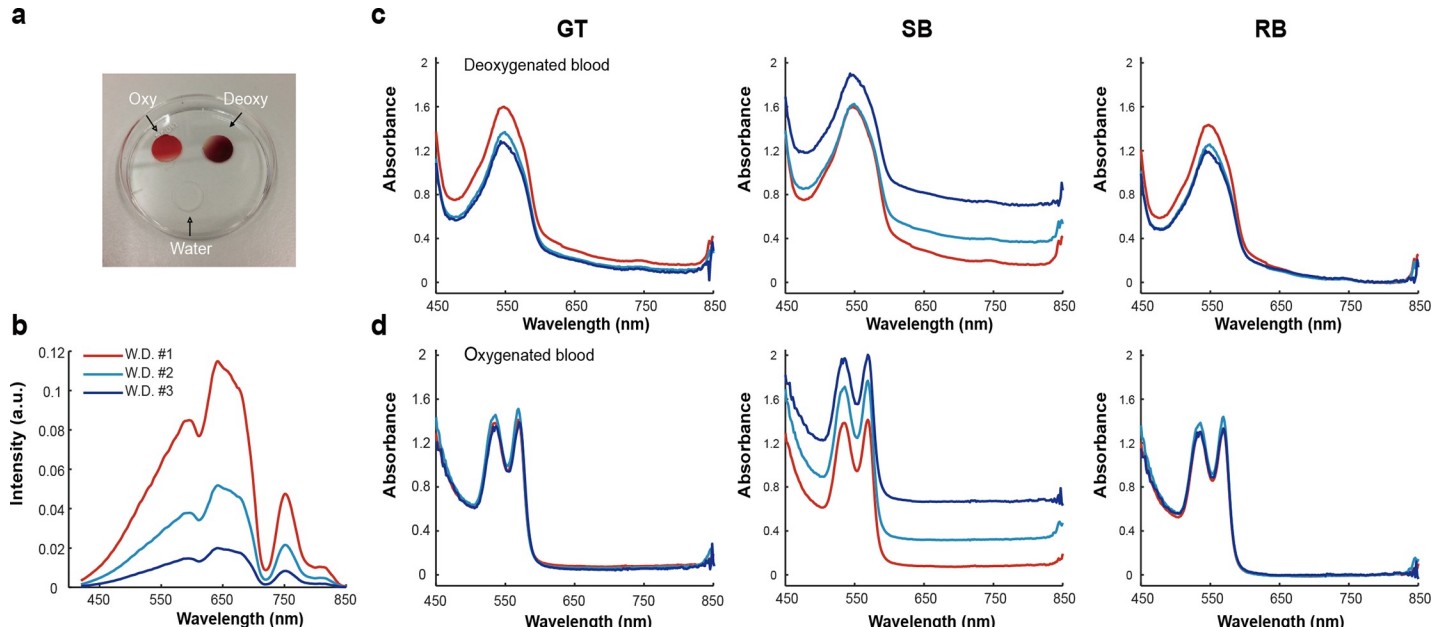

**Fig 3. Retrieved background (RB) signals enable accurate measurements of absorbance of deoxygenated and oxygenated blood compared to ground truth (GT).**
(a) Photograph of the experiment setup. Water (control) and blood (deoxygenation and oxygenation) were covered by a cover glass to prevent the sample from drying during the measurement. Hyperspectral imaging was performed using the internal illumination method at three working distances. (b) Experimentally measured reflectance signals of a control target (water) were used GT, with different optical illumination power according to working distance. Absorbance of deoxygenated (c) and oxygenated (d) blood measured at three working distances were then calculated using GT, SB and RB methods, showing good agreement between GT and RB, but substantial deviation for SB method.

methylene blue dyes added to test the effects of absorption and fluorescence, respectively (Fig 4A, see Methods). HySE was applied using internal illumination at 2 working distances; the spectral profile of agarose containing intralipid alone was used as GT (Fig 4B). Absorbance spectra of nigrosin and methylene blue calculated using the GT, SB and RB methods (Fig 4C and 4D) again show that GT and RB provide consistent spectral shapes, whereas SB has a substantial deviation in the profiles. The absorbance of nigrosin obtained using the retrieved BG is slightly lower in magnitude compared to GT, however, methylene blue is indistinguishable (0.83 ± 0.67% at peak 550 nm). This suggests that the underestimation observed in the blood and phantom experiments occurs because both haemoglobin and nigrosin have a small but finite absorption of light at 800 nm causing a slight inaccuracy in the BG estimation, whereas methylene blue has truly negligible absorption around 800 nm. Our RB method therefore leads to a slight underestimation of the actual absorbance values if light absorption around the chosen background wavelength is not negligible, though it does not change the absorbance spectrum itself.

## Application of the background correction method to biological tissue and endoscopic imaging conditions

To examine the practical application of the method, dissected chicken bone tissue, consisting of compact bone and bone marrow, was first measured (see Methods). A total of 150 spectral images of dissected chicken bone tissue were measured using the external illumination method to cast shadows across the topology of the sample, which can be seen in the synthetic RGB images (Fig 5A), created by the convolution of emulated RGB filters and measured hyperspectral signals (see Methods). Before examining the proposed method, the raw spectral signals in

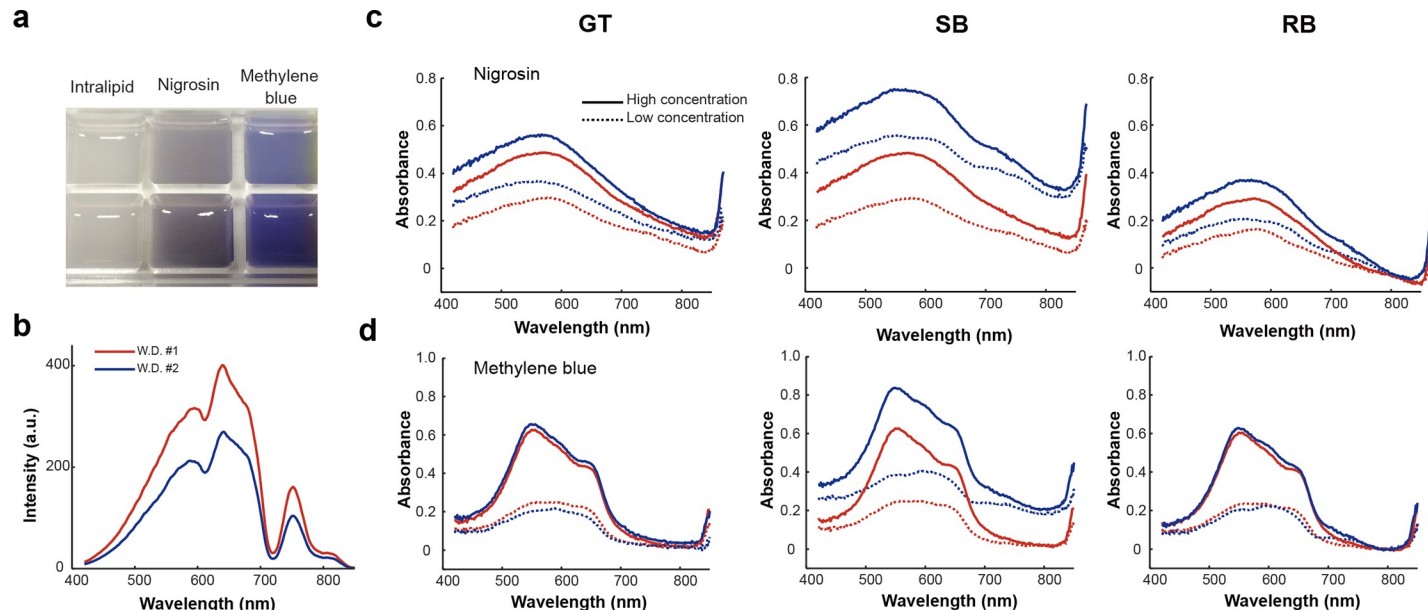

**Fig 4.** Background correction using the retrieved background method performs favourably in measurement of the absorbance spectra of nigrosin and methylene blue (a) Photograph of tissue-mimicking phantoms with intralipid (control), nigrosin (absorbing dye), and methylene blue (fluorescent dye). (b) Experimentally measured reflectance signals of the control phantom (intralipid only) at two different working distances gave the ground-truth background. (c) Absorbance of low and high concentrations of nigrosin dye calculated using the ground truth background (GT), single background (SB) and our retrieved background (RB) method respectively. (d) Absorbance of low and high concentrations of methylene blue calculated using GT, SB and RB methods respectively.

two square areas (4 by 4 pixels) within each of bone marrow, compact bone, and shadowed regions were assessed (S4A and S4B Fig). The reflected signal at the normalisation wavelength under the same illumination conditions should be similar in order to use the proposed method. The two nearby small squares in each tissue type were selected because illumination conditions in these small areas could be considered as homogenous. S4C Fig shows that raw reflected intensities at 800 nm of each tissue type are similar. There is no significant difference in the data recorded from the same tissue type.

The synthetic RGB images of GT and RB methods clearly show the structure of the tissue with uniform brightness, but the SB image shows bright and dark regions arising due to the uneven illumination (Fig 5A). Moreover, the shadowed region resulting from the sample morphology was restored to its original white colour only in the RB method. Representative absorbance images at three different wavelengths (456.1, 531.4 and 612.9 nm; Fig 5B) allow structures of the dissected chicken bone tissue to be visualised, showing qualitative similarity between GT and RB at all wavelengths, while the single BG reconstructions show different absorbance even in the same anatomical structures (solid and dashed white lines in Fig 5B).

In order to investigate spectral fidelity of the BG retrieval method, the average and standard deviation of the absorbance spectra in 6 different regions (red: bone marrow, orange: compact bone, and blue: white reflectance; indicated in Fig 5A) were quantified (Fig 5C). Spectral profiles of bone marrow and compact bone, in GT and RB show similar values and trends. In addition, our RB method brings the absorbance values on the left and right side of the white reflectance target closer compared to the result obtained by GT. The SB result, however, shows very different values and trends compared to the results obtained by the ground-truth and retrieved BG.

HySE was then applied in a tubular tissue-mimicking phantom with homogeneous methylene blue concentration (S5A Fig) placed on a tilted surface (S5B Fig). HySE was advanced

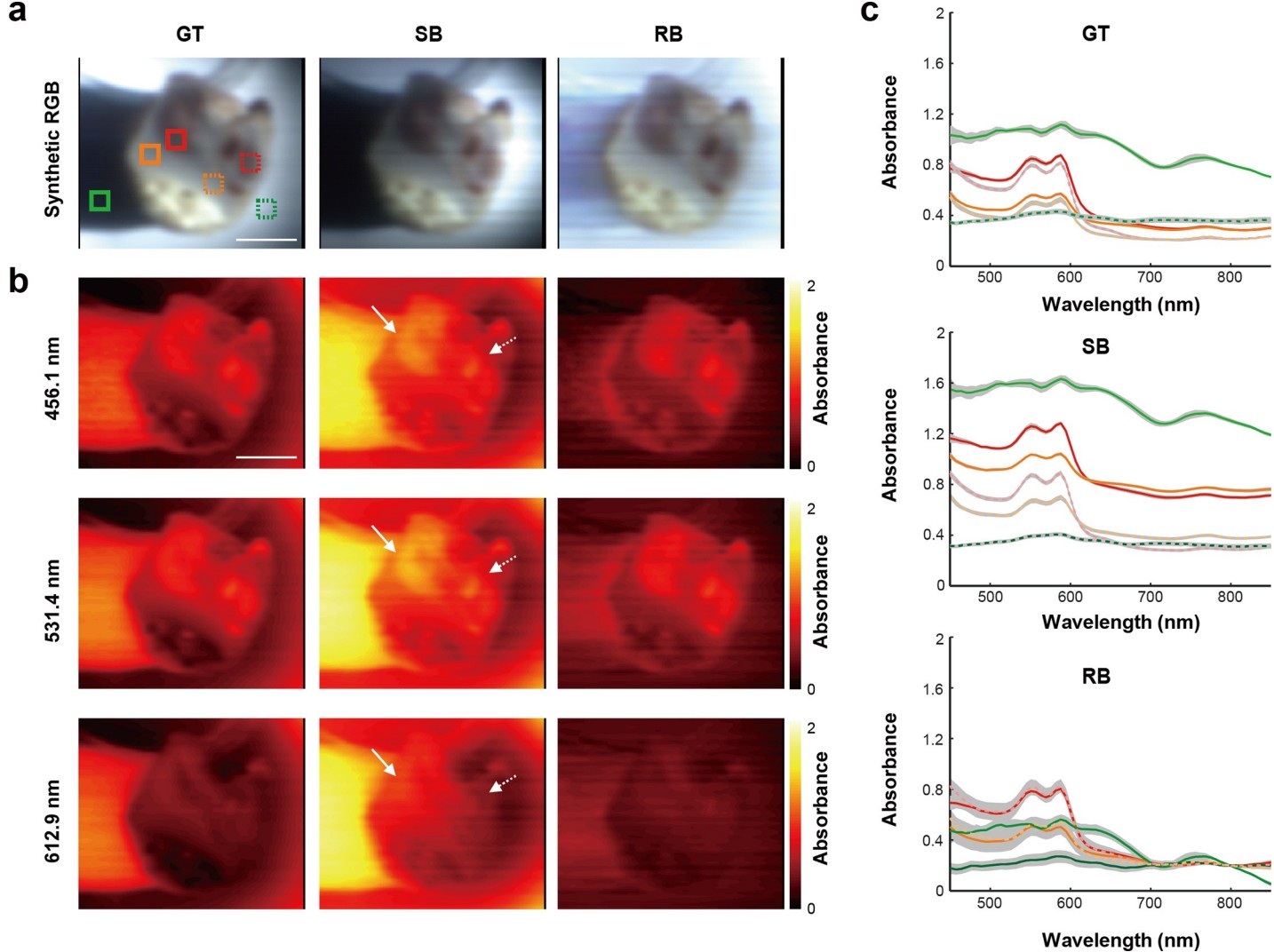

**Fig 5.** HSI data from a chicken tissue sample obtained using the RB method agrees well with the GT (a) Synthetic RGB images of the sample were created from hypercubes obtained by exploiting ground truth, single, and retrieved backgrounds. (b) Representative slice images from each hypercube (GT, SB and RB) were illustrated at three wavelengths (456.1, 531.4 and 612.9 nm). Solid and dashed arrows indicate anatomically similar structures in the sample. (c) Average absorbance of the hypercube reconstructed by using the GT, SB and RB methods within solid and dashed squares shown in (a) were obtained. Gray shaded area indicates the standard deviation. Scale bars: 1 cm.

horizontally into the tube with a motorised stage, which leads to a gradual decrease in the working distance of the endoscope. While the SB spectra show an offset as a function of working distance (S5C Fig), the RB spectra show the consistent measurement of absorbance regardless of the working distance (S5D Fig).

## Investigating the influence of background correction on hyperspectral data classification

To understand the extent to which incorrect background compensation influences HSI data classification, experimentally measured data obtained in the previous sections were composed into a set of 53 synthetic hypercubes in four steps: (1) generation of a random illumination pattern; (2) creation of a GT reflectance hypercube based on four experimentally measured signals

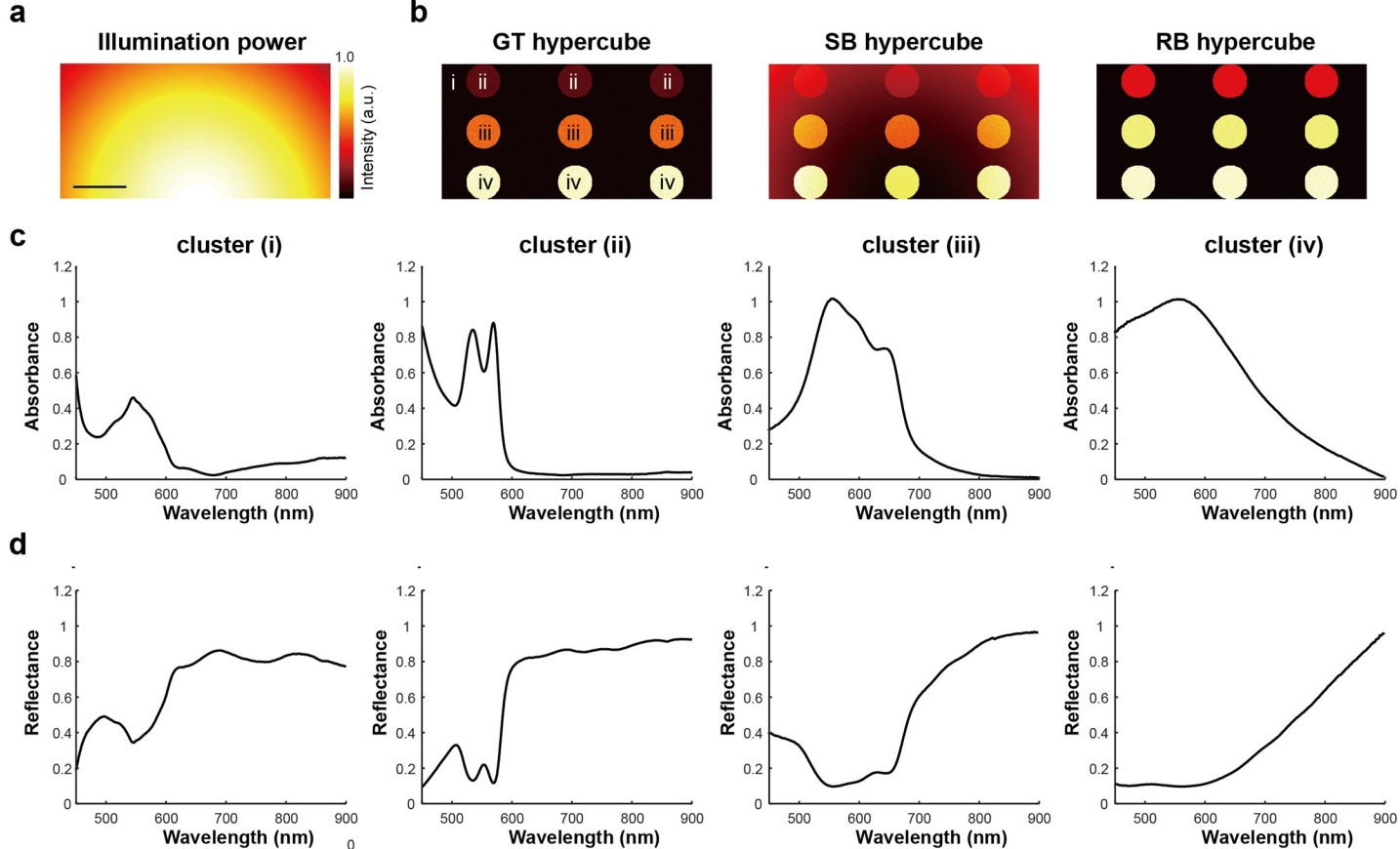

**Fig 6. Synthetic absorbance and reflectance hypercubes created based on experimentally measured hyperspectral signals and randomly generated illumination conditions. (a)** Representative image of Gaussian illumination power. **(b)** Representative projection images of synthetic GT, SB, and RB hypercubes. (i)-(iv) indicate the areas with spectral profiles of corresponding signals shown in (c, d). Three circles in the same horizontal position had the same spectral profile and were defined as a cluster. **(c, d)** Four experimentally measured absorbance and reflectance spectra (muscle tissue, oxygenated blood, methylene blue, and nigrosin samples) were exploited to create synthetic absorbance and reflectance hypercubes. Scale bar: 100 pixels.

(muscle tissue, oxygenated blood, methylene blue and nigrosine dyes) with an uncorrelated noise; (3) creation of SB and RB reflectance hypercubes by combining the GT hypercube with the random illumination pattern; and (4) applying a log-transformation of the produced reflectance hypercubes to generate absorbance hypercubes (Fig 6; see Methods).

The synthetic hypercubes were then subjected to PCA, SAM and machine learning classification. PCA is commonly used in HSI analysis for dimensionality reduction by finding a small number of orthonormal PCs that explain most of the variance of hyperspectral data, thus enabling simpler interpretation and classification. PCA was performed pixel-wise with singular value decomposition (SVD): hyperspectral data were centred by subtracting the mean values of each pixel from its corresponding signal, while the scaling (variance) was preserved due to the synthetic hypercubes being created under the same scale and unit conditions; the covariance matrix of the centred data was used as the SVD input. As the first and second PCs capture over 99% of the original variance, they were used to compare the influence of background correction methods. Scatter plots of PC2 against PC1 for GT, SB and RB in absorbance show no differences and 2D image of the scores on PC1 are also identical (S6A Fig). For reflectance, however, the scatter plots for SB show a dramatic elongation compared to the GT and RB methods and the 2D image of the SB PC1 scores clearly shows the power distribution of

illumination (S6B Fig), indicating an incomplete correction of the BG. Such behaviour arises because the incorrect BG causes scaling and shifting of the ground-truth reflectance and absorbance signals, respectively. Scaling changes the variance of the hyperspectral data, which produces an incorrect PCA result for the reflectance hypercube, whereas shifting of absorption data does not change PCA results as the variance is preserved.

SAM is widely used to evaluate the similarity between measured hyperspectral signals by calculating angles between them. Substantial differences in SAM analysis of the absorbance hypercube using the SB method were found compared to GT and RB and the SAM image again shows the power distribution of the illumination indicating an incomplete correction of the BG (S6C Fig). The reflectance data are identical regardless of the BG correction used (S6D Fig). This is because the scaling factor of the reflectance signal is eliminated through the calculation of the spectral angle in Eq (3) so it does not affect the SAM results, but shifting the absorption signal changes the calculated spectral angle values.

Finally, the effect of BG correction on machine learning-based data classification was evaluated through: classification based on the distance between the data and the centroid of each cluster by k-means clustering (K-Means, $k = 4$); maximising the distance between a decision boundary and members of different classes by support vector machines (SVMs); and training convolutional neural networks (CNNs). To enhance the learning process, min-max normalisation was employed with all three algorithms, to constrain the data between -1 and 1. For SVMs and CNNs, the supervised learning approach was employed with ground-truth data of 50 training hypercubes produced for each of the three BG correction methods, whereas K-Means was performed in an unsupervised learning manner without using data reduction methods such as PCA or SAM.

The test dataset was composed of three GT, SB, and RB hypercubes and the accuracy of all established classifiers was tested on all datasets (9 total comparisons). 100% classification accuracy is theoretically achievable due to the use of synthetic hypercubes, consisting of only four distinct spectral signals, for training and test. Using k-means clustering, the SB method showed accuracies of only 47.1% and 48.7%, respectively for the absorbance and reflectance hypercubes (Fig 7A and 7B) when clustered using the SB method classifier, compared to over 97.0%

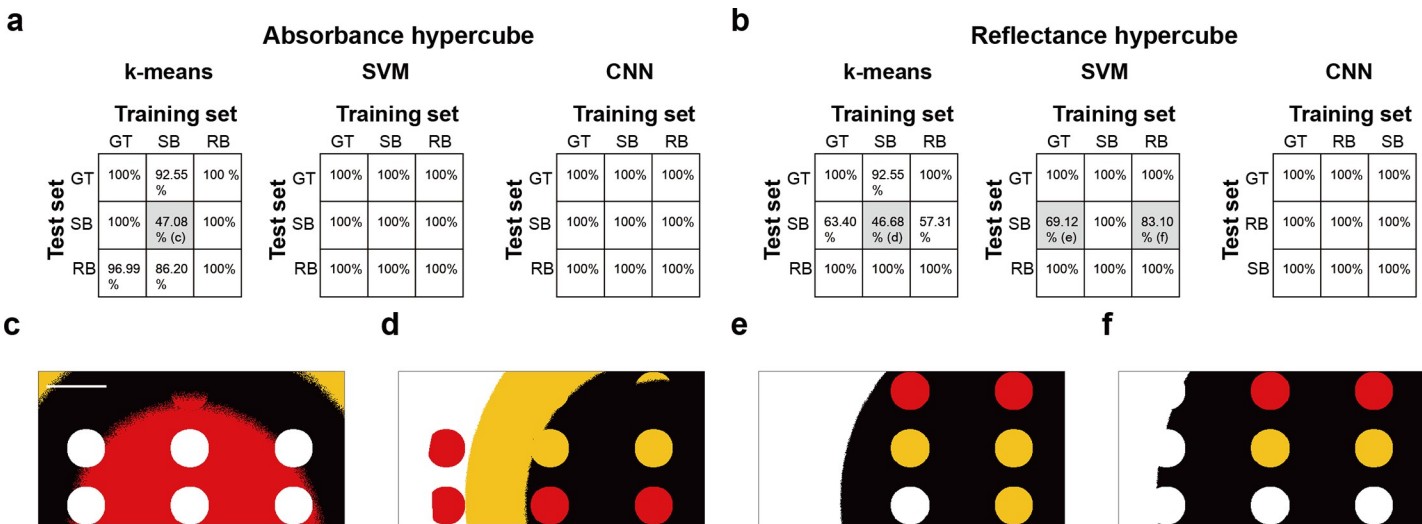

**Fig 7. Investigating the effect of background correction on the accuracy of machine learning-based hyperspectral imaging classification.** Classification accuracy of three machine learning methods (k-means clustering, support vector machine and convolutional neural network) in absorbance (a) and reflectance (b) hypercubes obtained by GT, SB and RB methods. (c—f) Representative images of classification results indicated by c–f in (a) and (b). Scale bar: 100 pixels.

for GT and RB method classifiers. The resulting segmented images again indicate incomplete background correction (Fig 7C and 7D) for the SB method. SVMs successfully segmented the four clusters in absorption hypercubes with 100% accuracy under all BG corrections, however, the classification accuracy of SB reflectance hypercubes segmented by using the SVMs trained via GT and RB hypercubes dropped to 69.1% and 89.1%, respectively again with incomplete background correction (Fig 7E and 7F). Lastly, CNNs were implemented via a six-layered network, including three convolutional layers, two fully connected layers and a softmax layer (S7 Fig). Trained CNNs classified the hypercubes with 100% accuracy regardless of hypercube types and BG conditions.

## Discussion

Applications of HSI in biomedicine frequently calculate optical reflectance and absorbance spectra for tissue classification. The data processing procedures assume that the samples are uniformly illuminated and while several methods can be employed to ensure that this assumption holds, applications that encounter variations in surface topology or optical power distribution, such as endoscopy, may result in classification errors. Here, we demonstrated a simple background correction method that enables estimation of the spectral profile and optical power distribution of illumination across a sample by exploiting the normalised spectra of the light source and intensity values of the measured hyperspectral signals at a fixed wavelength with negligible absorbance. The advantage of the method is that it is applied in software, so it does not require any specialised equipment or application of contrast agents and can be applied to any HSI data where a wavelength of negligible absorbance is available. It is therefore practical for application in biomedical imaging, for example, during hyperspectral endoscopy as demonstrated here using the HySE system. It could also be easily applied to snapshot multi-spectral biomedical imaging applications, if one of the wavelength bands is located in the NIR or other minimally absorbing wavelength range, which could enable the fast acquisition and online post-processing of the data.

We selected 800 nm as the wavelength for normalisation in these studies. The results suggest that in samples that are not absorbing at the selected normalisation wavelength, our retrieved background (RB) method accurately recovers the ground truth (GT) HSI data compared to the standard approach of using a single background (SB). The feasibility and applicability of the proposed RB method were demonstrated by measuring oxygenated and deoxygenated blood samples, a tissue-mimicking phantom with scattering, absorption, and fluorescence agents and *ex vivo* chicken tissue. These experiments indicated the importance of a complete background correction for analysis and interpretation of HSI data, with variations in optical power distribution causing rescaling of reflectance data and introducing offsets in absorbance data. Moreover, the importance of precisely retrieved and corrected background was assessed using HSI analysis methods and machine-learning based image classification techniques. In particular, the standard SB method led to erroneous findings for reflectance data in PCA and absorbance data in SAM. It also led to misclassification in both data types for k-means clustering and in reflectance data for SVMs, compromising their accuracy, however, well-trained CNNs were not vulnerable to changes in BG corrections or data types.

Nonetheless, there remain some limitations to the present study. The proposed method assumes that absorption at the normalisation wavelength is negligible. Should there arise some non-negligible or spatially inhomogeneous absorption at the normalisation wavelength, the calculated reflectance and absorbance may still introduce errors. For example, in our case using 800 nm as the normalisation wavelength, we saw that the blood samples and nigrosin dye samples had some non-negligible absorption at 800 nm, which meant that the magnitude

of the corrected spectrum could be up to 29.6% lower than the GT, but importantly, the shapes of the spectral profiles of the calculated reflectance and absorbance remained unaffected. Therefore, prior information about the absorbance of a given sample at the normalisation wavelength is necessary when making comparisons between the magnitude of the recorded spectra. While many biological tissues have little absorption at 800 nm [40,41] choosing this wavelength may produce problems for experiments that introduce NIR dyes for molecular imaging. Selecting a wavelength further into NIR tissue optical window could overcome this, though would require illumination of the tissue with further NIR/IR optical power and the associated thermal deposition characteristics should be carefully considered from a safety perspective.

In addition, we examined the influence of noise on the study and found that the accuracy of the normalisation method decreases with increasing noise in the spectra. Care should therefore be taken when applying the method to a noisy spectral data set. Another consideration is the need for spectrally uniform illumination across the target, which is an important precondition for many experiments in HSI and also affects the proposed method. If multiple incoherent light sources are used, then spectral homogeneity should be checked before using the proposed method. A further consideration is that the effects of BG correction on HSI classification using machine learning algorithms were tested here using simple synthetic hypercubes composed of experimentally measured data from only four spectra components. While these serve to illustrate the potential of the method in cases where known ground truth is available, further experiments would be needed to establish the bounds of operation of the method in another chosen application. Finally, we focused on the influence of background correction on reflectance and absorbance hypercubes. Further work would be needed to understand how well the method could perform for other HSI applications, such as multiplexing of fluorescence contrast agents [12].

Despite these limitations, the proposed background correction method allows for accurate and consistent measurement of HSI data, regardless of illumination method and optical power distribution. Application of the method could facilitate further exploitation of multi- and hyperspectral imaging techniques in practical clinical applications, where controlling the illumination pattern and power are non-trivial.

## Supporting information

**S1 Fig. Optical design of the line-scanning hyperspectral endoscope.** The system is assembled using a CE-marked endoscope with an imaging fibre bundle and an integrated illumination fibre. A sample is illuminated either by coupling a halogen light source to the illumination fibre (internal illumination method) or by directly illuminating via the fibre-coupled halogen light source (external illumination method). Hyperspectral data is acquired using a CCD coupled to the spectrograph. For line-scanning hyperspectral imaging, a motorized translational stage is exploited to control imaging position in these studies. Abbreviations: CCD, charge coupled device; L1–2, lens; Obj1–2, objective lens.
(PNG)

**S2 Fig. Schematic for obtaining the ground truth background (GT), single background (SB) and retrieved background (RB). (a)** GT was obtained by measuring a white reflectance target under the same position and illumination conditions as the sample measurement. From the GT, the normalised spectral profile of the background was calculated by averaging across all spatial locations within the hyperspectral image frame. One of GTs was used as SB. **(b)** To obtain RB, the intensity ratio ($C_s/C_b$) at 800 nm and the normalised spectral profile of the background was calculated. The intensity ratio of each vertical pixel was calculated by dividing

intensity values of a sample spectral image ($C_s$, red dashed line) at 800 nm by the intensity value of normalised background signal at 800 nm ($C_b$). **(c)** The spectrum of the RB used for correction at a specific vertical pixel was determined by multiplying the normalised background to the intensity ratio value corresponding to the pixel.
(PNG)

**S3 Fig.** Influence of noise-to-signal ratio to the retrieved BG method (a) Simulation of raw spectral profiles of oxygenated blood with different signal-to-noise ratios (1%, 5%, 10%, and 20%). (b) Absorbance obtained using spectral signals in (a) and the retrieved BG method. Gray shaded area indicates the standard deviation. (c). Bar graphs show the average error percentages of absorbance at four different signal-to-noise ratios. Error bar indicates the standard deviation.
(PNG)

**S4 Fig.** Investigation of intensity variation at 800 nm (a) Left: Synthetic RGB image of chicken tissue. Right: Magnified image of the dashed square shown in left figure. Scale bars: 1 cm (b) Average measured spectral profiles of the bone marrow (BM), compact bone (C), and shade (S) areas within solid squares shown in (a) were obtained. Gray shaded area indicates the standard deviation. (c). Bar graphs show average intensities of six regions shown in (a) were calculated. Error bar indicates the standard deviation. Statistical analysis was performed using Student t-test.
(PNG)

**S5 Fig. The RB method enables the accurate measurement of absorbance in endoscopy conditions.** (a) Photograph of the tubular tissue-mimicking phantom with homogeneous methylene blue concentration. (b) Schematic of the experiment. Absorbance of the tissue-mimicking phantom at three working distances was obtained using SB (c) and RB methods (d). The solid line and the gray shaded area indicate average absorbance and standard deviation, respectively.
(PNG)

**S6 Fig. Assessment of background effects on hyperspectral image analysis via principal component analysis (PCA) and spectral angle mapping (SAM). (a, b)** Scatter plots of 2nd principal component (PC) versus 1st PC (top) and representative images of 1st PC scores (bottom) of absorbance and reflectance hypercubes, respectively. Scale bar: 100 pixels. **(c, d)** Bar graphs indicate mean and standard deviation (error bars) of angle values for each cluster shown in the bottom image (top) and 2D images of spectral angle values (bottom) of absorbance and reflectance hypercubes, respectively. SAM was performed using the average spectral profile of the cluster *i* of each hypercube. Scale bar is 100 pixels.
(PNG)

**S7 Fig. Schematic process for the application of convolutional neuronal networks.**
(PNG)

## Acknowledgments

We thank Dr Laura Bollepalli for technical assistance in the completion of this study.

## Author Contributions

**Conceptualization:** Jonghee Yoon, Sarah E. Bohndiek.

**Data curation:** Jonghee Yoon.

**Formal analysis:** Jonghee Yoon, Alexandru Grigoroiu.

**Funding acquisition:** Sarah E. Bohndiek.

**Investigation:** Jonghee Yoon.

**Methodology:** Jonghee Yoon.

**Project administration:** Jonghee Yoon.

**Resources:** Sarah E. Bohndiek.

**Software:** Jonghee Yoon, Alexandru Grigoroiu.

**Supervision:** Jonghee Yoon, Sarah E. Bohndiek.

**Validation:** Alexandru Grigoroiu, Sarah E. Bohndiek.

**Writing – original draft:** Jonghee Yoon, Sarah E. Bohndiek.

**Writing – review & editing:** Jonghee Yoon, Sarah E. Bohndiek.

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
