## [Decision Letter · Decision Letter 0]

19 Nov 2019

PONE-D-19-28216

A background correction method to compensate illumination variation in hyperspectral imaging

PLOS ONE

Dear Dr Bohndiek,

Thank you for submitting your manuscript to PLOS ONE. After careful consideration, we feel that it has merit but does not fully meet PLOS ONE’s publication criteria as it currently stands. Therefore, we invite you to submit a revised version of the manuscript that addresses the points raised during the review process.

We would appreciate receiving your revised manuscript by Jan 03 2020 11:59PM. To enhance the reproducibility of your results, we recommend that if applicable you deposit your laboratory protocols in protocols.io, where a protocol can be assigned its own identifier (DOI) such that it can be cited independently in the future. For instructions see: http://journals.plos.org/plosone/s/submission-guidelines#loc-laboratory-protocols

We look forward to receiving your revised manuscript.

Kind regards,

Ireneusz Grulkowski, PhD

Academic Editor

PLOS ONE

Journal Requirements:

1. We note that you have stated that you will provide repository information for your data at acceptance. Should your manuscript be accepted for publication, we will hold it until you provide the relevant accession numbers or DOIs necessary to access your data. If you wish to make changes to your Data Availability statement, please describe these changes in your cover letter and we will update your Data Availability statement to reflect the information you provide.

2. Please do not include funding sources in the Acknowledgments or anywhere else in the manuscript file. Funding information should only be entered in the financial disclosure section of the submission system. https://journals.plos.org/plosone/s/submission-guidelines#loc-acknowledgments

3. Please amend the Methods section of your manuscript to include the source of the animal tissues

Additional Editor Comments (if provided):

Dear Authors,

Please, see the attached reviews. Please, revise the manuscript by addressing the comments given by the reviewers.

Reviewers' comments:

Reviewer's Responses to Questions

**Comments to the Author**

1. Is the manuscript technically sound, and do the data support the conclusions?

Reviewer #1: Yes

Reviewer #2: Partly

Reviewer #3: Yes

Reviewer #4: Yes

2. Has the statistical analysis been performed appropriately and rigorously? 

Reviewer #1: Yes

Reviewer #2: Yes

Reviewer #3: Yes

Reviewer #4: Yes

3. Have the authors made all data underlying the findings in their manuscript fully available?

Reviewer #1: Yes

Reviewer #2: Yes

Reviewer #3: Yes

Reviewer #4: Yes

4. Is the manuscript presented in an intelligible fashion and written in standard English?

Reviewer #1: Yes

Reviewer #2: Yes

Reviewer #3: Yes

Reviewer #4: Yes

5. Review Comments to the Author

Reviewer #1: In this paper, the authors propose a simple method that is able to compensate the effects of illumination variations across a sample in hyperspectral imaging applications, using information about a specific spectral band in which the sample has approximately zero absorbance, and the spectral profile of the illumination source. This avoids the need for specialized illumination equipment.

The problem is important and well motivated (i.e. poorly compensated illumination affects subsequent image analysis tasks), and the method is simple but seems robust as long as the hypotheses are satisfied. Nonetheless, there are some points that could be improved.

- The description of the proposed method in page 5 could be made clearer if supported by mathematical equations. More precisely, the effect of the illumination scaling alpha in equations 1 and 2, the relationship between C_b and C_s and the quantities introduced in equations 1 and 2, and the result of the proposed background correction could be shown in a few equations without substantially increasing the section's length.

Also, note that alpha is used to denote both the illumination scaling and the spectral angle mapper result in equation 3, what could cause some confusion.

- A non-zero absorbance of the sample at the 800nm spectral band is shown to lead to a slight underestimation of the amplitude of the spectral samples. It also seems to be assumed that this absorbance is the same (i.e. theoretically zero) for all pixels in the sample. What effect would be observed in the proposed method if there are (e.g. small) spatial variations of this quantity? To which extent is this expected to occur in the application of interest? It would be interesting to see a discussion on this matter.

- How is the pixel spectra reshaped in figure S.5 into a 2D signal?

Reviewer #2: Here, my personal review to the Authors for the article PONE-D-19-28216:

In first place I would like to thanks the authors. Despite the complexity of the topic of the paper, I found the paper well written and clear. Moreover, an interesting method to correct for illumination inhomogeneities directly form the measured sample is presented and tested in several ex-vivo conditions. Besides of this, here are some suggestions and comments that can enhance paper robustness.

Major comments:

1) Figure. 1: a confront is made between normalized and not normalized spectrum between the retrieved illumination and the measured illumination. It would be clearer if the comparison would be made on the same graph to show the consistency in between the spectral signals. I suggest to place both dashed and continue lines for the measured and the retrieved data, respectively. This comment applies to all the data where the differences between methods are minimal such as in the GT-RB comparisons in Figures 2-5.

2) It is not clear to me why the author selects 800 nm as reference wavelength. Only reference 38 is given to support this choice. In my opinion a detailed explanation of the choice of 800 nm as reference wavelength must be provided. Taking under consideration the used sample and the measurements performed. This will help also to understand how this method could be used for different samples where the wavelengths that must be selected would be different.

3) If I did not misunderstand the text, the information at 800 nm is completely lost no matter what the sample. If it is the case add a statement on this in the text

4) In Fig 5 all the information after 750 nm is cut. Please insert all the spectral information to confront these spectra with all the others.

5) Which is the influence of using 800 nm to correct these images in terms of artifacts? I propose a study of the variation of the intensity at very small area where the illumination light can be considered perfectly homogenous and independent from the illumination conditions in the case of the chicken breast where the sample is not perfectly homogeneous in reflectance. This validation will grant more robustness to the overall article. Is not clear to me the variations of spectral information at 800 nm in its absolute value. If the differences are considerable the method proposed will produce unpredictable artifacts. Maybe the Root Mean Square value could be calculated in between standard deviations calculated from the data collected on different areas (marrow and compact bone). In this way the effect of the 800 nm choice can be estimated.

6) “Despite these limitations, the proposed background correction method allows for accurate and consistent measurement of HSI data, regardless of illumination methods and optical power distribution.”

This is partially true as one of the assumptions is that every spectral channel must be acquired in the same illumination condition. In other words that means a good superposition between spectral channels. That is the case of most of the applications when the spectral sampling is performed in the detection, but it is not true in the case that several light sources are used instead.

7) In this case an a-priori study of the sample is necessary at least to select the right wavelength. I suggest to add a statement that underline this. In fact, if the sample is highly etherogenous in its composition, with high variations for all the wavelength in the spectral range this method cannot be blind as it will gives an unpredictable result in terms of spectral information.

Minor comments:

8)The “hypercube is a n-dimensional volume of data for a 3D hyperpestral cube, in my advise it would be better datacube”

Reviewer #3: This manuscript proposed a background correction method to compensate for the variations in surface morphology or light power distribution at the sample. The proposed background correction method enables the estimation of optical characteristics of illumination at the sample, which is a major advance in hyperspectral endoscopy systems. The authors experimentally demonstrated the feasibility of the proposed background correction method using hyperspectral imaging data acquired via a hyperspectral endoscopy system from different samples. The proposed method and experiment results are useful in further exploitation of hyperspectral imaging endoscopy systems in practical clinical applications, the manuscript can be accepted for publication.

A few questions that should be addressed:

1. Where is the multiplicative factor alpha in eq. (1) and constant in eq. (2) ?

2. In the Result section P5, line 10. Did you multiply the normalized spectrum of the light source with the intensity ratio between the measured spectral profiles of the light source and the sample or with the intensity ratio between the normalized spectral profiles of light source and sample?

3. The wavelength 800 nm was selected as the band with low absorption to verify the feasibility of the proposed method. The question is, if the absorption of a certain tissue is not that small at 800nm, does the method still work? If not, how to solve this problem?

4. In figure 5a, there is no much difference in darkness between RGB images of GT and SB? In contrast, GT and SB looks similar.

5. In figure 5a, six colored rectangular boxes are obscure.

Reviewer #4: The article by Yoon et al. presents a relatively simple approach to correct intensity variations in sample illumination in HSI applications. Overall, the presented approach is scientifically sound and offers an easy correction option with some assumptions. However, the simplicity of the approach means it is not innovative in general, but rather as an application to a specific field. There a few major points to be addressed:

1) Sensitivity to noise – a method calculating a scaling coefficient only using a single wavelength will be more susceptible to noise over methods using multiple. Since most of the work is using some sort of simulation, it would be interesting to see the influence of varying levels of noise on the retrieved signal. Please add such calculations.

2) Compare to other baseline correction methods (e.g. independent component analysis, fitting) and show the benefits of this approach. It may be in the lack of optimization parameters and speed.

3) The machine learning part in general is valid, however, a small detail would make a big difference. The authors did a min/max scaling for the SB dataset, while anyone doing spectroscopy and machine learning would also remove the baseline in any way before doing that. Subtract minimum value or a linear fit (or any other baseline remocal) for SB and then do classification for KNN and SVM.

A smaller point is to slightly clear out the introduction about absorbance and reflectance with scaling factor, since in a proper spectroscopic experiment the scaling would also affect background and be removed in the proportion of I/I0. The lack of proper background in these experiments is the problem.

There are also a few small language inaccuracies here and there.

6. PLOS authors have the option to publish the peer review history of their article (what does this mean?). If published, this will include your full peer review and any attached files.

Reviewer #1: No

Reviewer #2: No

Reviewer #3: Yes: Tingkui Mu

Reviewer #4: No

---

## [Author Response · Author response to Decision Letter 0]

17 Jan 2020

The response to reviewer and editor comments has been uploaded separately.

---

## [Decision Letter · Decision Letter 1]

4 Feb 2020

PONE-D-19-28216R1

A background correction method to compensate illumination variation in hyperspectral imaging

PLOS ONE

Dear Dr Bohndiek,

Thank you for submitting your manuscript to PLOS ONE. After careful consideration, we feel that it has merit but does not fully meet PLOS ONE’s publication criteria as it currently stands. Therefore, we invite you to submit a revised version of the manuscript that addresses the points raised during the review process.

We would appreciate receiving your revised manuscript by Mar 20 2020 11:59PM. To enhance the reproducibility of your results, we recommend that if applicable you deposit your laboratory protocols in protocols.io, where a protocol can be assigned its own identifier (DOI) such that it can be cited independently in the future. For instructions see: http://journals.plos.org/plosone/s/submission-guidelines#loc-laboratory-protocols

We look forward to receiving your revised manuscript.

Kind regards,

Ireneusz Grulkowski, PhD

Academic Editor

PLOS ONE

Additional Editor Comments (if provided):

Please, address short comments of the reviewer #3. Then the paper will be ready for publishing.

Reviewers' comments:

Reviewer's Responses to Questions

**Comments to the Author**

1. If the authors have adequately addressed your comments raised in a previous round of review and you feel that this manuscript is now acceptable for publication, you may indicate that here to bypass the “Comments to the Author” section, enter your conflict of interest statement in the “Confidential to Editor” section, and submit your "Accept" recommendation.

Reviewer #1: All comments have been addressed

Reviewer #3: All comments have been addressed

Reviewer #4: All comments have been addressed

2. Is the manuscript technically sound, and do the data support the conclusions?

Reviewer #1: Yes

Reviewer #3: Yes

Reviewer #4: Yes

3. Has the statistical analysis been performed appropriately and rigorously? 

Reviewer #1: Yes

Reviewer #3: Yes

Reviewer #4: Yes

4. Have the authors made all data underlying the findings in their manuscript fully available?

Reviewer #1: Yes

Reviewer #3: Yes

Reviewer #4: Yes

5. Is the manuscript presented in an intelligible fashion and written in standard English?

Reviewer #1: Yes

Reviewer #3: Yes

Reviewer #4: Yes

6. Review Comments to the Author

Reviewer #1: (No Response)

Reviewer #3: The authors made important revision to the manuscript and answered all the questions very carefully. It is obvious that the authors added math to describe the background correction method in the revised manuscript, which greatly increased the readability of the paper. In addition, in the discussion part of the revised manuscript, the authors pointed out the limitations of the method comprehensively, which give more clear direction for future research and application. In conclusion, I recommend this manuscript for publication.

There are some points that could be improved:

1. In P12, the word “measured” in the sentence “We propose instead to multiply the normalized spectrum of the light source (Fig. 1e) with the intensity ratio between the measured spectral profiles of the light source (Cb) and the sample (Cs) at a wavelength of low absorbance in the sample to estimate the actual spectrum of the light source at the target (Fig. 1f)” may need to be changed as “normalized”.

2. In the equations 6 and 7, the denominator and numerator are all functions of (x,y), so it is better to change Cs to Cs(x,y).

Reviewer #4: (No Response)

7. PLOS authors have the option to publish the peer review history of their article (what does this mean?). If published, this will include your full peer review and any attached files.

Reviewer #1: No

Reviewer #3: No

Reviewer #4: No

---

## [Author Response · Author response to Decision Letter 1]

7 Feb 2020

Reviewer #1: (Thank you for your review. We are pleased the revised manuscript was satisfactory.)

Reviewer #3: The authors made important revision to the manuscript and answered all the questions very carefully. It is obvious that the authors added math to describe the background correction method in the revised manuscript, which greatly increased the readability of the paper. In addition, in the discussion part of the revised manuscript, the authors pointed out the limitations of the method comprehensively, which give more clear direction for future research and application. In conclusion, I recommend this manuscript for publication.

There are some points that could be improved:

1. In P12, the word “measured” in the sentence “We propose instead to multiply the normalized spectrum of the light source (Fig. 1e) with the intensity ratio between the measured spectral profiles of the light source (Cb) and the sample (Cs) at a wavelength of low absorbance in the sample to estimate the actual spectrum of the light source at the target (Fig. 1f)” may need to be changed as “normalized”.

2. In the equations 6 and 7, the denominator and numerator are all functions of (x,y), so it is better to change Cs to Cs(x,y).

We would like to thank the reviewer for their time in reviewing our manuscript. We have revised the manuscript according to reviewer’s comments. 

Reviewer #4: (Thank you for your review. We are pleased the revised manuscript was satisfactory.)

---

## [Decision Letter · Decision Letter 2]

10 Feb 2020

A background correction method to compensate illumination variation in hyperspectral imaging

PONE-D-19-28216R2

Dear Dr. Bohndiek,

We are pleased to inform you that your manuscript has been judged scientifically suitable for publication and will be formally accepted for publication once it complies with all outstanding technical requirements.

With kind regards,

Ireneusz Grulkowski, PhD

Academic Editor

PLOS ONE

Additional Editor Comments (optional):

Reviewers' comments:

Reviewer's Responses to Questions

**Comments to the Author**

1. If the authors have adequately addressed your comments raised in a previous round of review and you feel that this manuscript is now acceptable for publication, you may indicate that here to bypass the “Comments to the Author” section, enter your conflict of interest statement in the “Confidential to Editor” section, and submit your "Accept" recommendation.

Reviewer #3: All comments have been addressed

2. Is the manuscript technically sound, and do the data support the conclusions?

Reviewer #3: Yes

3. Has the statistical analysis been performed appropriately and rigorously? 

Reviewer #3: Yes

4. Have the authors made all data underlying the findings in their manuscript fully available?

Reviewer #3: Yes

5. Is the manuscript presented in an intelligible fashion and written in standard English?

Reviewer #3: Yes

6. Review Comments to the Author

Reviewer #3: (No Response)

7. PLOS authors have the option to publish the peer review history of their article (what does this mean?). If published, this will include your full peer review and any attached files.

Reviewer #3: Yes: Tingkui Mu

---

## [Editor Report · Acceptance letter]

27 Feb 2020

PONE-D-19-28216R2 

A background correction method to compensate illumination variation in hyperspectral imaging 

Dear Dr. Bohndiek:

I am pleased to inform you that your manuscript has been deemed suitable for publication in PLOS ONE. Congratulations! Your manuscript is now with our production department. 

With kind regards,

on behalf of

Dr. Ireneusz Grulkowski 

Academic Editor

PLOS ONE